**Data Availability Statement:** All relevant data are contained within the manuscript and its Supporting Information files, or available on the Challenge

# Crowdsourced identification of multi-target kinase inhibitors for RET- and TAU- based disease: The Multi-Targeting Drug DREAM Challenge

Zhaoping Xiong[1‡], Minji Jeon[2‡], Robert J. Allaway[3‡], Jaewoo Kang[2,4], Donghyeon Park[2], Jinhyuk Lee[2], Hwisang Jeon[4,5], Miyoung Ko[2], Hualiang Jiang[1,5], Mingyue Zheng[5], Aik Choon Tan[6], Xindi Guo[3], The Multi-Targeting Drug DREAM Challenge Community[¶], Kristen K. Dang[3], Alex Tropsha[7], Chana Hecht[8], Tirtha K. Das[8], Heather A. Carlson[9], Ruben Abagyan[10], Justin Guinney[3], Avner Schlessinger[11]*, Ross Cagan[8,12]*

1 Shanghai Institute for Advanced Immunochemical Studies, ShanghaiTech University, Shanghai, China, 2 Department of Computer Science and Engineering, Korea University, Seoul, Republic of Korea, 3 Sage Bionetworks, Seattle, Washington, United States of America, 4 Interdisciplinary Graduate Program in Bioinformatics, Korea University, Seoul, Republic of Korea, 5 Drug Discovery and Design Center, State Key Laboratory of Drug Research, Shanghai Institute of Materia Medica, Chinese Academy of Sciences, Shanghai, China, 6 Department of Biostatistics and Bioinformatics, Moffitt Cancer Center, Tampa, Florida, United States of America, 7 Laboratory for Molecular Modeling, Division of Chemical Biology and Medicinal Chemistry, UNC Eshelman School of Pharmacy, University of North Carolina, Chapel Hill, North Carolina, United States of America, 8 Department of Cell, Developmental, and Regenerative Biology, Icahn School of Medicine at Mount Sinai, New York City, New York, United States of America, 9 Department of Medicinal Chemistry, University of Michigan, Ann Arbor, Michigan, United States of America, 10 Skaggs School of Pharmacy and Pharmaceutical Sciences, University of California, San Diego, California, United States of America, 11 Department of Pharmacological Sciences, Icahn School of Medicine at Mount Sinai, New York City, New York, United States of America, 12 Institute of Cancer Sciences, University of Glasgow; Glasgow, Scotland, United Kingdom

‡ These authors share first authorship on this work.
¶ Membership of The Multi-Targeting Drug DREAM Challenge Community is provided in the Acknowledgments.
* avner.schlessinger@mssm.edu (AS); ross.cagan@glasgow.ac.uk (RC)

## Abstract

A continuing challenge in modern medicine is the identification of safer and more efficacious drugs. Precision therapeutics, which have one molecular target, have been long promised to be safer and more effective than traditional therapies. This approach has proven to be challenging for multiple reasons including lack of efficacy, rapidly acquired drug resistance, and narrow patient eligibility criteria. An alternative approach is the development of drugs that address the overall disease network by targeting multiple biological targets ('polypharmacology'). Rational development of these molecules will require improved methods for predicting single chemical structures that target multiple drug targets. To address this need, we developed the Multi-Targeting Drug DREAM Challenge, in which we challenged participants to predict single chemical entities that target pro-targets but avoid anti-targets for two unrelated diseases: RET-based tumors and a common form of inherited Tauopathy. Here, we report the results of this DREAM Challenge and the development of two neural network-

landing page (www.doi.org/10.7303/syn8404040). Code and Data availability Zhaoping Xiong's top-performing method is available in a Github repository: https://github.com/xiongzhp/FusedEmbedding. The final embedding data used for this method are available at: https://raw.githubusercontent.com/xiongzhp/FusedEmbedding/master/data/kinase_seq_embedding.csv Instructions for running the method can be found here: https://github.com/xiongzhp/FusedEmbedding/blob/master/REPRODUCING.md. Team DMIS-MTD's top-performing method is available at: https://github.com/dmis-lab/ReSimNet https://www.synapse.org/#!Synapse:syn20545440 Instructions for running this method can be found at: https://github.com/dmis-lab/ReSimNet/blob/master/README.md. Chemical fingerprints and final embedding data used by team DMIS-MTD are available on the Challenge workspace: https://www.synapse.org/#!Synapse:syn25702146. A detailed description of the implementation and results of these two methods as well as all other submitted methods can be found in the Supplemental Methods. Post challenge analysis notebooks can be retrieved from: https://github.com/Sage-Bionetworks-Challenges/Multi-Target-Challenge.

**Funding:** RC received funding from the National Institutes of Health, grant: U54OD020353 from the Office of Research Infrastructure Programs (https://orip.nih.gov/). RA was funded by the National Institute of General Medical Sciences grant: R35GM131881 (https://www.nigms.nih.gov/) The funders had no role in study design, data collection and analysis, decision to publish, or preparation of the manuscript.

**Competing interests:** I have read the journal's policy and the authors of this manuscript have the following competing interests: Avner Schlessinger discloses that he is a co-founder of Alchemy, LLC. Ross Cagan discloses that he receives compensation as a consultant to Vivan.

based machine learning approaches that were applied to the challenge of rational polypharmacology. Together, these platforms provide a potentially useful first step towards developing lead therapeutic compounds that address disease complexity through rational polypharmacology.

## Author summary

Many modern drugs are developed with the goal of modulating a single cellular pathway or target. However, many drugs are, in fact, 'dirty;' they target multiple cellular pathways or targets. This phenomenon is known as multi-targeting or polypharmacology. While some strive to develop 'cleaner' therapeutics that eliminate secondary targets, recent work has shown that multi-targeting therapeutics have key advantages for a variety of diseases. However, while multi-targeting drugs that affect a precisely-defined profile of targets may be more effective, it is difficult to computationally predict which molecules have desirable target profiles. Here, we report the results of a competitive crowdsourcing project (the Multi-Targeting Drug DREAM Challenge), where we challenged participants to predict chemicals that have desired target profiles for cancer and neurodegenerative disease.

This is a *PLOS Computational Biology* Benchmarking paper.

## Introduction

Despite important advances in drug development, many diseases remain partly or wholly resistant to drug-based treatments. In recent decades, the field has attempted to address this by developing precision therapeutics with the goal of targeting critical nodes in disease networks. However, this approach has proven to be challenging. Most targeted therapeutics do not progress past preclinical research or clinical trials due to poor efficacy or unacceptable toxicity [1,2]. Additional hurdles are encountered even after clinical approval. For example, initial efficacy against melanoma by the BRAF inhibitors dabrafenib or vemurafenib as single-agent therapeutics is generally followed by emergent tumor resistance [3]. Further, based on biomarkers only a small number of patients with metastatic tumors are eligible for these target-driven precision therapies, and fewer still show sustained response [4].

An alternative to developing drugs with single targets is the development of polypharmacology (multi-targeting) drugs. Many clinically approved drugs bind multiple targets (http://ruben.ucsd.edu/dnet) [5,6], and in some cases the drug is improved by these secondary activities. For example, vandetanib, a drug used in RET-dependent medullary thyroid cancer (MTC), has a broad range of kinase targets that include RET, VEGFR, etc. [7]; pre-clinical studies suggest that some of these secondary targets can contribute to drug efficacy (*e.g.*, [8,9]). Toxicity is also an important consideration in multi-targeting drug activity. For example, imatinib's side effects—including inhibition of proper bone remodeling—are likely due to multiple 'off-targets' of imatinib [10]. Developing optimized polypharmacology-based drugs remains a challenge as there are few roadmaps for predicting compounds that safely combine beneficial targets ('pro-targets') while avoiding 'anti-targets' (direct targets that are liabilities in the context of a specific assay) in a single molecule.

Current approaches to predict multi-target compounds range from 'physics-based' methods that directly predict binding affinities, to 'omics-based' approaches utilizing large datasets such as those including side-effects or gene expression profiles [11–15]. Notably, recent advances in machine learning architectures have allowed the development of more effective methods relying on both physics-based and omics-based approaches [16,17]. For example, a generative tensorial reinforcement learning (GENTRL) model was used to design a small-molecule with optimal biological activity for the discoidin domain receptor 1 (DDR1) with desirable pharmacokinetic properties in just 21 days [18]. However, several limitations remain: current methods often (i) identify frequent binders or obvious compounds, (ii) fail to discriminate activities among close analogs, or (iii) identify small molecule inhibitors of 'promiscuous kinases' (e.g., DDR1 [19]) that are inherently easier to target. Notably, exploration of the full chemical space—estimated between $10^{24}$ and $10^{60}$ structures—still remains a tremendous challenge: known binders of, *e.g.*, the kinome together address a limited set of chemical scaffolds [20].

Rational polypharmacology will require an improved ability to predict chemical structures that specifically target optimized target sets. To identify better computational strategies for this problem, we developed the Multi-Targeting Drug DREAM Challenge [21]. In this challenge, we challenged participants to predict single chemical entities that target pro-targets but avoid anti-targets for two unrelated diseases: RET-based tumors and an inherited form of Tauopathy. Exploring each disease, we used a Drosophila 'dominant genetic modifier screen' to identify mediators of Drosophila $RET^{M955T}$-mediated transformation and of Tau-mediated dysfunction, two well-characterized models of human disease [22,23]. Although Drosophila-based screening platforms provide an imperfect model of human disease, we have previously used them to help identify effective pharmacologic interventions for RET-driven, colorectal, and adenoid cystic cancers [8,24,25]. The resulting functional suppressors and enhancers of RET-driven transformation provide a useful 'roadmap' towards an ideal profile of a lead polypharmacological therapeutic.

We used data from our dominant genetic modifier screens to identify suppressors (candidate 'pro-targets') and enhancers (candidate 'anti-targets') for RET- and TAU-based disease. We publicly posted this data, challenging the community to develop computational approaches with improved ability to predict single chemical structures that successfully inhibit pro-targets while avoiding anti-targets. Here, we report the development of two artificial neural network-based machine learning approaches that were applied to the challenge of rational polypharmacology. Together, these platforms provide a potentially useful first step towards developing lead therapeutic compounds that address disease complexity through polypharmacology.

## Results

Rational development of drugs that act through polypharmacology will require identifying (i) pro-targets that, as a group, are predicted to be effective in addressing a disease network, and (ii) anti-targets that are predicted to act as liabilities either due to reduced disease efficacy or increased whole body toxicity. Our work in Drosophila [22,23,26–30] identified whole body pro-targets and anti-targets for RET-associated cancer and Tauopathy. To leverage this data, we established the Multi-targeting Drug DREAM challenge (www.doi.org/10.7303/syn8404040) designed to promote and assess novel rational approaches to polypharmacology (Fig 1).

### Pro-targets and anti-targets

To ensure whole animal relevance, we used data from two Drosophila models. To model RET-dependent oncogenic transformation, we used data from previous work that screened Drosophila $GMR$-$RET^{M955T}$ and $ptc$>$RET^{M955T}$ cancer models [22,23]. Broad genetic screening

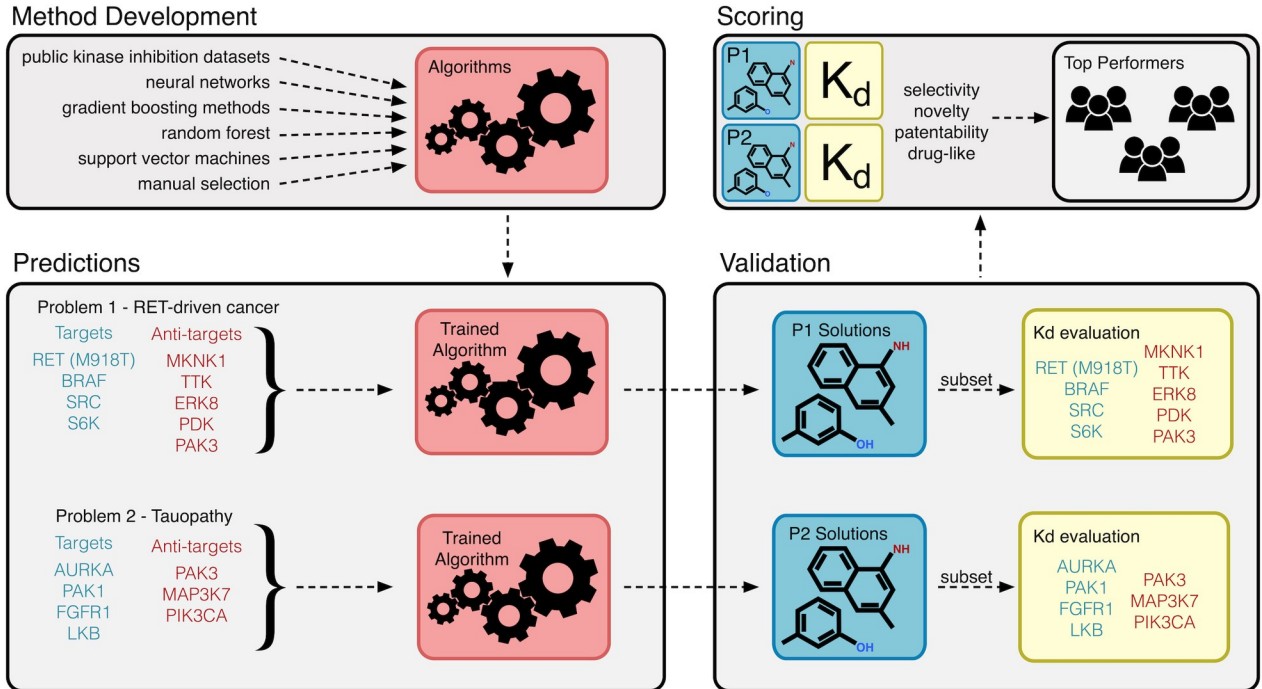

**Fig 1. Multi-targeting drug DREAM Challenge overview.** This challenge sought to identify new algorithms for the identification of multi-targeting drugs that bind disease-relevant targets and avoid disease-specific anti-targets. Participating teams used a broad array of different datasets and computational approaches to develop these methods. They were then asked to use these methods to predict small molecules that would bind to RET-driven cancer targets and anti-targets (Problem 1, P1) as well as Tauopathy targets and ant-targets (Problem 2, P2). A shortlist of these predicted molecules were then experimentally validated (Kd determination) as binders or non-binders of P1 and P2 proteins. Finally, the submitted compounds were scored using a combination of criteria including selectivity, novelty, patentability, and drug-like properties to identify the top performing teams.

identified multiple key 'pro-targets' defined as 'dominant genetic suppressors' of $GMR\text{-}RET^{M955T}$-mediated transformation including Drosophila orthologs of BRAF, SRC, and S6K. Candidate 'anti-targets'—genetic loci that, when reduced, acted as 'dominant genetic enhancers' of $RET^{M955T}$-mediated transformation—include MKNK1, TTK, ERK8, PDK1, and PAK3. Given its central role, we also included RET as a pro-target.

To provide a second independent test for compound predictions, we used data recently generated in a screen of a previously-described inherited Drosophila Tauopathy model [26–30]. Expressing the human disease isoform $TAU^{R406W}$ exogenously in a transgenic Drosophila line led to nervous system defects and animal lethality [26–30]. Genetic screening of most of the Drosophila kinome identified several dominant genetic suppressors of $TAU^{R406W}$. For example, reducing the activity of the Drosophila orthologs of AURKA, PAK1, FGFR1, or STK11 suppressed $TAU^{R406W}$-mediated fly lethality; these loci represent candidate therapeutic targets. Conversely, reducing activity of PAK3, MAP3K7, or PIK3CA enhanced $TAU^{R406W}$-mediated lethality, indicating they should be avoided by candidate therapeutics.

We used these two sets of data as benchmarks for computation-based compound predictions: ideal therapeutic leads would inhibit most or all pro-targets while exhibiting minimum activity against anti-targets.

## Participation and performance

The challenge had 190 registered participants. Of these registrants, 34 submitted predictions, either as a part of one of 5 teams or as individuals. 10 prediction sets were submitted for

**Table 1. Scoring guidelines for Problems 1 and 2.** Starred targets indicate that binding or avoiding that target was a requirement to receive target-based points.

| Criteria | Points Awarded |
|---|---|
| *Problem 1* | |
| Binds RET[M918T)] | 5 |
| Binds BRAF, SRC, S6K | 1, 3 or 9 if binding 1, 2 or 3 of these targets |
| Avoids MKNK1* | 3 |
| Avoids TTK, ERK8, PDK, PAK3 | 1, 2, 3, or 4 if it avoids 1, 2, 3, or 4 of these targets |
| *Problem 2* | |
| Binds AURKA* | 5 |
| Binds PAK1* | 5 |
| Binds FGFR1, LKB | 1 or 3 if binding 1 or 2 of these targets |
| Avoids PAK3* | 3 |
| Avoids MAP3K7* | 3 |
| Avoids PIK3CA | 1 |
| Predicted to enter CNS | 3 |
| *Both Problems* | |
| Novelty | 2 |
| Ability to Patent | 2 |
| Drug-like molecule | 3 |

Problem 1, and 8 prediction sets were submitted for Problem 2. Each prediction was scored using the rubric described in Table 1 (see Materials and Methods for additional details).

## Problem 1: Prediction of multi-targeting compounds for RET-driven cancers

Problem 1 asked participants to predict molecules that bound the RET$^{M955T}$-associated cancer pro-targets RET[M918T], BRAF, SRC, S6K, and did not bind the anti-targets MKNK1, TTK, ERK8, PDK1, and PAK3. Of the 10 prediction files submitted, two of the predictions had the same top-ranked molecule and, therefore, 9 compounds were selected by the challenge organizers for experimental evaluation. S1A Fig shows that while most of the submitted compounds had no similar molecule submitted by another team (Tanimoto similarity >0.5), the few similar submitted molecules were mostly predicted by the same method/team.

Binding evaluation revealed that the majority of predicted compounds (7/9) did not bind any of the pro-targets at 10 μM or any of the anti-targets at 30 μM (S1 Table). Two compounds were observed to significantly bind one or more pro-targets: ZINC98209221 and ZINC40900273 (Figs 2 and 3). ZINC98209221 was observed to significantly bind all 4 pro-targets, while avoiding 2 of the 5 anti-targets. Of the remaining 3 'bound' anti-targets, the K$_d$ was larger than that of the pro-targets, indicating that ZINC98209221 displayed a lower binding affinity for the Problem 1 anti-targets as compared to the pro-targets. The other compound in Fig 3, ZINC40900273, only significantly bound one of 4 pro-targets (BRAF) but avoided all 5 anti-targets.

## Problem 2: Prediction of multi-targeting compounds for Tauopathy

Problem 2 asked participants to predict compounds that bind the Tauopathy pro-targets AURKA, PAK1, FGFR1, and STK11 and avoid the Tauopathy anti-targets PAK3, MAP3K7, and PIK3CA. Eight compounds were selected for experimental testing. Kinase binding assays indicated that 6 of 8 of the predicted compounds did not bind any of the pro-targets or anti-

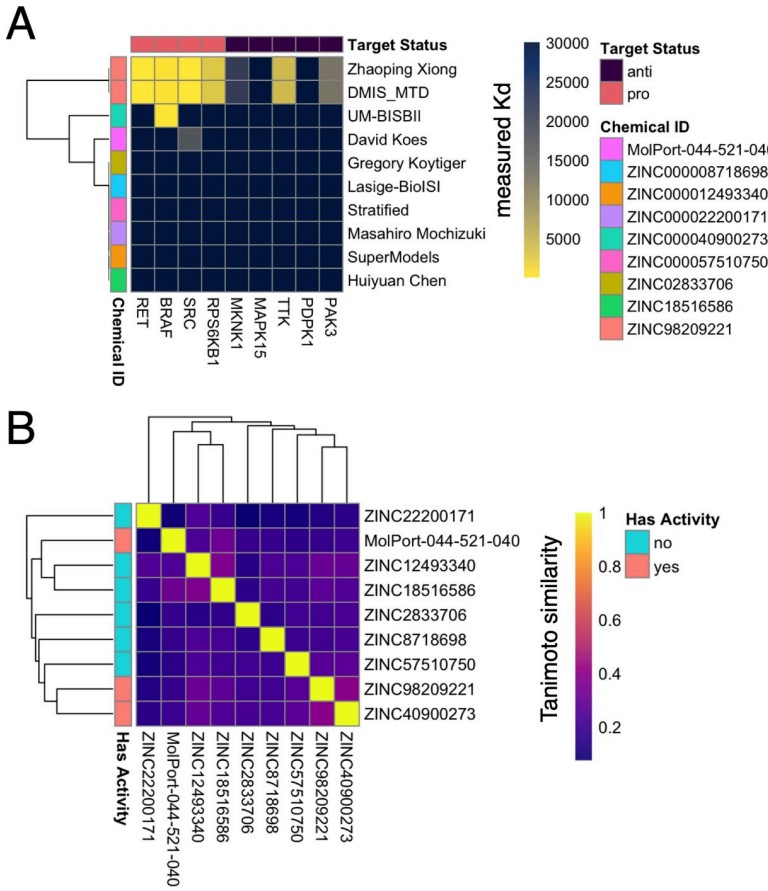

**Fig 2. Kd values of experimentally tested candidates from Problem 1.** A heatmap indicating the $K_d$s of all compounds tested in Problem 1. The best binder of the pro-targets (ZINC98209221) was predicted by two different teams.

targets at 30 μM (S2 Table). Two compounds were observed to bind one or more pro-targets: ZINC3938668 and ZINC538658 (Figs 3 and 4). S1B Fig recapitulates the finding from Problem 1 in Problem 2—most of the submitted compounds had no similar molecule submitted (Tanimoto similarity >0.5) and when similar molecules were submitted, they were generally predicted within a single method and not across different teams.

The molecule predicted by Zhaoping Xiong, ZINC3938668, bound 3 of 4 pro-targets at 10 μM: AURKA, PAK1, and STK11. However, this molecule also bound 2 of 3 anti-targets at 10 μM: PAK3 and MAP3K7. Similarly, the predicted molecule from team DMIS-MTD, ZINC538658, bound 3 of 4 pro-targets at 10 μM: AURKA, FGFR1, and STK11, and 2 of 3 anti-targets at 10 μM: PAK3 and MAP3K7.

## Overall performance and methods summaries

To identify the top participants, we used the scoring algorithm described in Table 1. This algorithm considered multiple factors including binding of targets and avoiding anti-targets as well as fulfilling novelty, patentability, and drug-likeness criteria to derive a numerical score. The top teams for both Problem 1 and Problem 2 were identical: Zhaoping Xiong and DMIS_MTD, with 16 points each for Problem 1, and 11 and 9 points for Problem 2, respectively (Table 2). The remaining teams received points for novelty, patentability, and drug-

**Fig 3. Compounds with activity against Problem 1, 2 targets.** "Submitter or Team" indicates the team that identified the compound as a solution to Problem 1 (top) or Problem 2 (bottom) as indicated. "Chemical ID" is the ZINC identifier for the molecule. "Structure" is a two-dimensional representation of the molecule. "InChIKey" is the hashed InChI for the molecule. Strikingly, two teams identified the same molecule as a solution for Problem 1.

likeness criteria, but the compounds predicted by these teams failed to meet binding criteria to receive points. Therefore overall top-performers for the challenge were Zhaoping Xiong and DMIS_MTD, with overall scores of 28 and 25, respectively (Table 2).

Across the 10 participating teams a variety of approaches were employed to arrive at a set of predicted compounds for Problems 1 and 2 (Table 3 and S1 Methods). The majority of these

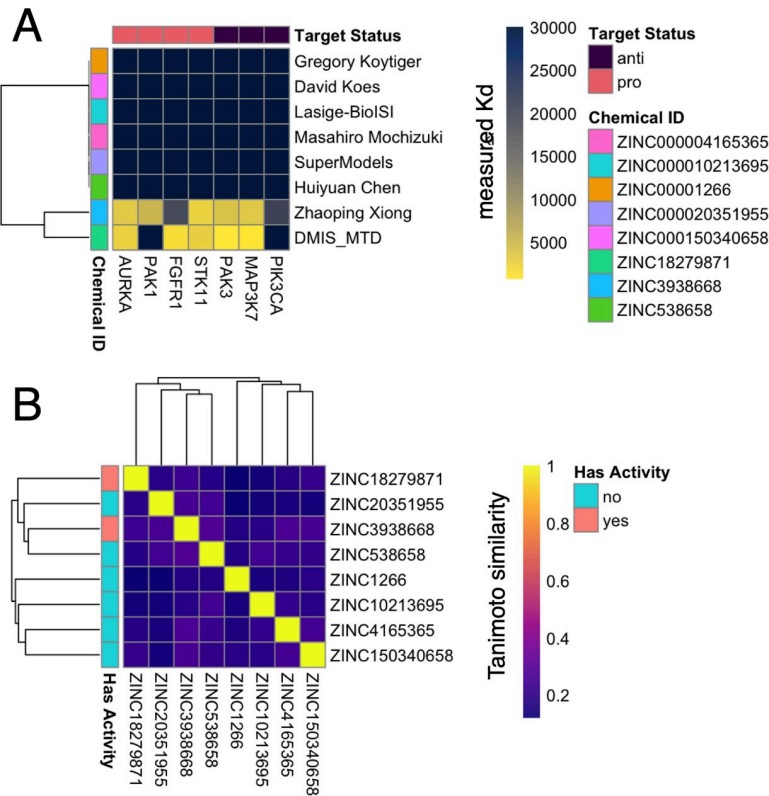

**Fig 4. Kd values of experimentally tested candidates from Problem 2.** A heatmap indicating the $K_d$s of all compounds tested in Problem 1. ZINC3938668 and ZINC538658 bind a subset of the pro-targets, and also both bind 2 of the three anti-targets.

methods utilized multiple steps including (i) combining structure- and ligand-based approaches, (ii) applying one or more machine learning/classification algorithms, (iii) use of other computational approaches such as filtering, ligand docking, or creation and application of confidence scoring metrics to identify compounds for the two problems, or (iv) manual selection of the top candidates. Importantly, some teams used fully automated methods, while others used approaches that required several manual curation steps. While the majority of

**Table 2. Challenge scoreboard.** "Submitter or Team" indicates which team was scored. The "Problem 1" and "Problem 2" scores indicate each team's performance in the two problems as scored using the criteria described in Table 1. "Interactome" indicates whether the team participated in a text-mining exercise to facilitate team interaction for bonus points. "Final Points" indicates the final score for each team.

| Submitter or Team | Problem 1 | Problem 2 | Interactome | Final Points |
|---|---|---|---|---|
| Zhaoping Xiong | 16 | 11 | TRUE | 28 |
| DMIS_MTD | 16 | 9 | FALSE | 25 |
| Lasige-BioISI—Multi-target Drug Designing | 7 | 7 | TRUE | 15 |
| SuperModels | 7 | 7 | FALSE | 14 |
| Huiyuan Chen | 7 | 7 | FALSE | 14 |
| Masahiro Mochizuki | 7 | 7 | FALSE | 14 |
| David Koes | 7 | 7 | FALSE | 14 |
| Gregory Koytiger | 7 | 7 | FALSE | 14 |
| UM-BISBII | 8 | | FALSE | 8 |
| Stratified | 7 | | FALSE | 7 |

**Table 3. Methods summary table.** "Computational Approach Used" describes the methods that were mentioned in each team's writeup (see S1 Methods). "Manual Selection of Top Candidates" indicates whether a team described using a manual curation step to produce their final set of predicted compounds. "Method Data Description" highlights the data sources and data types used by each team.

| Team | Computational Approach Used | Manual Selection of Top Candidates | Method Data Description |
|---|---|---|---|
| David Koes | neural network | yes | PDB |
| Team Stratified | ensemble gradient boosting | yes | pIC50 |
| DMIS-MTD | neural network | yes | Cmap scores |
| UM-BISBII | dataset filtering | | SEA-TC algorithm |
| Lasige-BioISI—Multi-target Drug Designing | random forest, support vector machine | yes | ChEMBL |
| Gregory Koytiger | neural network | | ChEMBL, SwissProt |
| SuperModels | random forest, logistic regression | yes | ChEMBL, Klaeger et al, Eidogen, ExCAPE-DB |
| Zhaoping Xiong | neural network | yes | Merget and Fulle et al, Uniprot |
| Masahiro Mochizuki | random forest, logistic regression | | unspecified kinase inhibition assay dataset |
| Huiyuan Chen | similarity calculations | | Drugbank |

teams utilized one or more machine learning approaches, the two top-performing teams specifically relied on neural network-based approaches to identify the best compounds for Problem 1 and 2 (Fig 3). We therefore examined these two approaches more closely to identify key similarities and differences.

## Top performing teams

The two teams that were most effective at predicting binding of pro-targets and avoidance of anti-targets both used artificial neural network approaches. Method 1 (Zhaoping Xiong) used features derived from the protein and small molecules. Method 2 (Team DMIS_MTD) relied on transcriptional responses in combination with chemical features of the drug, with a post processing step that included docking.

Method 1 used a graph convolutional neural network deep learning approach [31,32] to predict candidate bioactive compounds (Fig 5). Ligand-based models are constrained when testing is done with just a few bioactive compounds. To build a more generalizable and useful model, this approach incorporated key features of each target kinase by converting the amino acid sequence into a 'word embedding' vector of a length of 80, then trained together with graph convolutional neural fingerprints which includes 4 layers of GRUs (Gated Recurrent Units) as the aggregating method. In developing this method, different combinations were used of extended connectivity fingerprints (ECFP4) of compounds with known bioactivity data, neural fingerprints, and protein embeddings. Testing on a 10% internal hold-out test set demonstrated that the combination of neural fingerprints and protein embeddings, which we have termed FEMTD (Fused Embedding for Multi-Targeting Drug), performed the best, with an RMSE of 0.414 (S3 Table).

The Method 1 researcher also made the observation that the majority of $-\log_{10}$ IC$_{50}$ (pIC50) training data was centered around a pIC50 of 5, making prediction of more extreme compound-target pIC50 values more challenging. To address this, the method was developed as a classification problem in which the bioactivity of training data was classified as "inactive" (pIC50 < 5), "weak" (5 < = pIC50 < 6) or "potent" (pIC50 > = 6). A classification model trained on these data was then used to predict compound classes for the targets/anti-targets posed by the Challenge. Finally, the top predictions were manually selected based on predicted

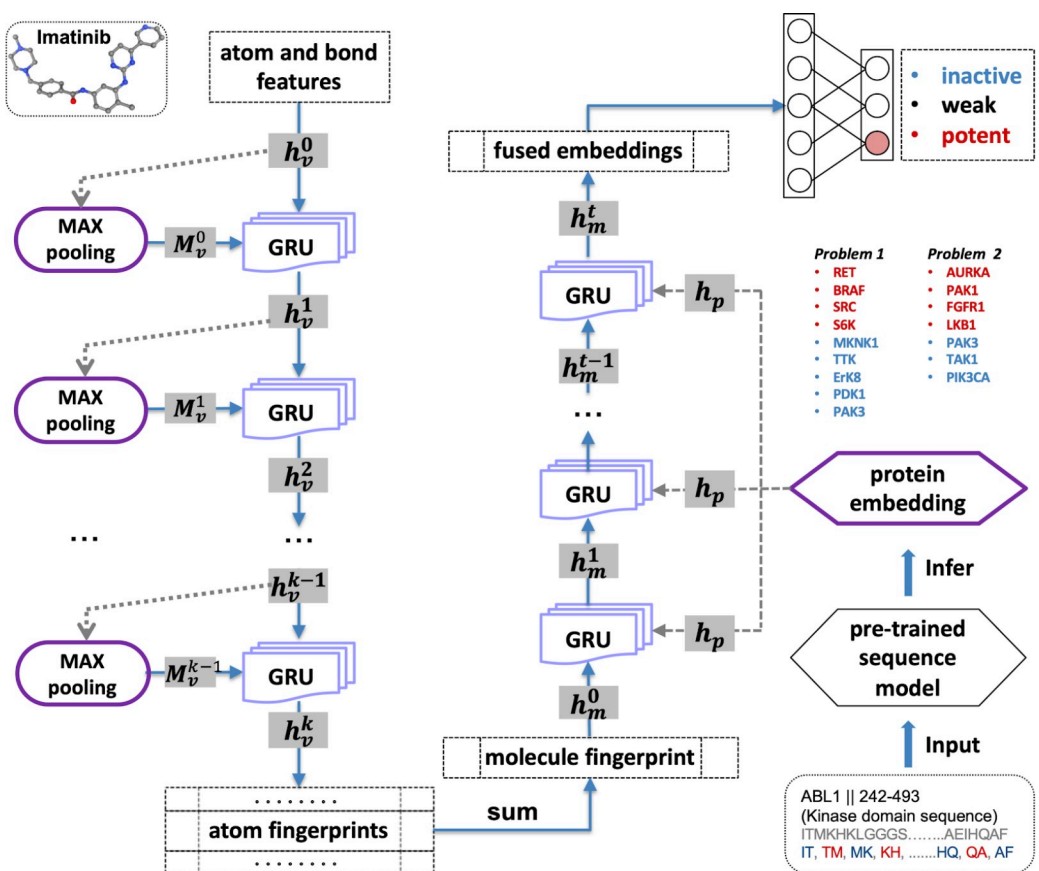

**Fig 5. A schematic summarizing FEMTD (Zhaoping Xiong's method).** The molecule is embedded with graph neural fingerprints and the target proteins are represented as sequences and pretrained with a language model (doc2vec) model. These two embeddings are fused together to classify the binding affinity as inactive, weak, or potent.

classes for the query targets and anti-targets. Overall, this model performed best in the Challenge (S3 Fig).

Team DMIS-MTD based Method 2 on ReSimNet [33], a Siamese neural network model that receives the structural information of two drugs represented as 2048-bit extended connectivity fingerprints (ECFPs) and predicts their transcriptional response similarity. ReSimNet was trained on the Connectivity Map (CMap) dataset [34], which provides CMap scores of around 260,000 drug pairs from 2,400 unique drugs. The CMap score of a pair of drugs is computed based on the similarity of the observed drug-induced gene expression profiles of the two drugs. Trained on these CMap scores of observed drug pairs, ReSimNet predicts the CMap scores of any drug pair that includes novel compounds. ReSimNet consists of two identical 2-layer Multi-Layer Perceptrons (MLPs), and the dimension of a hidden layer and an output layer is 512 and 300, respectively. The cosine similarity of two output vectors is considered predicted CMap scores of two compounds. The model was trained to minimize the mean squared error between the real CMap scores and the predicted CMap scores of compound pairs using Adam optimizer [35] with a learning rate of 0.005. The input fingerprints and the output embedding vectors of the CMap compounds are available on the Challenge workspace (see Code and Data availability).

Before utilizing ReSimNet to identify novel drug candidates, Team DMIS-MTD first identified several prototype drugs with kinase inhibition profiles that satisfied most of the

**Fig 6. A schematic summarizing DMIS-MTD's method.** The method was based on ReSimNet [33], a Siamese neural network model that predicts their transcriptional response similarity of two compounds. (A) Several prototype drugs whose kinase inhibition profiles satisfy most of the pro-target and anti-target conditions of the challenge problem were identified using the KInase Experiment Omnibus (KIEO) database. (B-C) ReSimNet, trained on the CMap dataset, was used to predict the transcriptional response similarity between a prototype drug and a novel compound from ZINC15. (D) Finally, the top-ranking ZINC15 compounds are aggregated and filtered for compounds that fulfill Lipinski's rules, a patent search through PubChem, and a protein-ligand docking algorithm.

Challenge's pro-target and anti-target conditions. They used the kinase inhibition profiles from the *KInase Experiment Omnibus* (KIEO) database (http://kieo.tanlab.org), which was constructed from experimental data on kinase inhibitors. The data was curated from more than 600 published articles (Fig 6A). ReSimNet, trained on the CMap dataset (Fig 6B), was used to predict the transcriptional response similarity between a prototype drug and a novel compound from ZINC15 [36] (Fig 6C). Finally, the top-ranking ZINC15 compounds were aggregated and filtered by Lipinski's rule, a patent search through PubChem, and a protein-ligand docking algorithm (Fig 6D).

### Translation of hits to in vivo model systems

After identifying the best-binders in the competitive phase of the Challenge, we assessed the *in vivo* translatability of the *in silico* predicted binders from Problem 1 that were also found to bind the P1 targets *in vitro* (Figs 2A and 3). We obtained and tested the compounds for their ability to rescue viability of transgenic flies that expressed the oncogenic form of Drosophila RET in multiple tissues ($ptc>RET^{M955T}$) as previously described [23,37].

Control transgenic $ptc>RET^{M955T}$ (RET2B) flies fail to eclose to adulthood due to expression of the transgene (0% adult viability; S4 Fig and S4 Table). Feeding $ptc>RET^{M955T}$ flies with food containing AD80 or APS6-45 (50 μM)—experimental multi-targeting kinase inhibitors that reduce $ptc>RET^{M955T}$ lethality [23,37]—significantly increased survival to 27% and 22% mean eclosure, respectively (S4 Fig and S4 Table). Feeding with ZINC98209221 or ZINC40900273 (1, 10, or 50 μM final food concentrations) rescued to 1–1.5% of animals, respectively (S4 Fig and S4 Table), a number that failed to rise to statistical significance.

### Ensemble modeling

As a proof-of-concept to identify candidates for future study, we devised a "wisdom of the crowds" approach to identify consensus Problem 1 predictions of the top-performing models by identifying the nexus of chemical space predicted by these methods. Specifically, we obtained a larger list of top predictions from the two top-performing teams using the same method and then determined the pairwise Tanimoto similarity (S2A Fig) between these two sets. Using this data, we performed network analysis (S2B Fig), which allowed us to identify subclusters of structurally similar predicted molecules within the full set of top predicted molecules. We further profiled these molecules using computationally generated absorption,

distribution, metabolism and excretion (ADME) metrics (S5 Table) to facilitate future prioritization of these molecules for medicinal chemistry and *in vivo* screening.

## Discussion

One approach to overcoming the limitations of precision medicine approaches in drug development is to develop new drugs with a goal of multi-targeting drugs that target multiple nodes in a disease network. Developing these drugs is a challenge, as both desired targets as well as toxic targets must be considered in the design process. One possible framework for identifying these drugs involves the use of predictive computational modeling strategies to perform virtual screening and identify compounds with a desirable target profile. Here, we described our efforts to identify better computational approaches for this pharmacologic problem.

To provide a disease context to this challenge, we used a set of previously identified pro-targets and anti-targets to design two challenge problems. Problem 1 asked teams to predict molecules that bound the RET[M955T]-associated cancer targets RET[M918T], BRAF, SRC, S6K, and avoided binding to MKNK1, TTK, ERK8, PDK1, and PAK3. Problem 2 asked participants to predict compounds that bound the Tauopathy pro-targets, AURKA, PAK1, FGFR1, and STK11 and did not bind PAK3, MAP3K7, and PIK3CA. The teams collectively identified 3 compounds for Problem 1 and 2 compounds for Problem 2 that were observed to have any binding potency to the pro-targets. Furthermore, the same set of teams in both problems were successful, indicating that their methods were consistently best-performing. Both of the top-performing teams used neural networks based learning approaches. However, there were other teams that also employed a neural network based approach but did not successfully identify any binding compounds, suggesting that the specific approach may be critical for success. Other approaches (solely manual curation, random forest, logistic regression) were unsuccessful in identifying binders, suggesting that these methods may not be well-poised to perform well in this type of problem. In the future, it may prove valuable to explore how additional high-dimensionality training data types—for example, high-content image-based screening [38] or pharmacogenomic datasets [34,39,40]—can improve performance or enable the community to explore different prediction strategies and methodologies.

Another key finding was that most of the submitted methods, including the top-performing methods, required a manual curation/selection step to select the top performing compounds (Table 3). The decisions and criteria that are used in manual selection can be difficult to define or reproduce using a decision tree or other algorithm. A clear direction for future studies is to better define the nature and rationale for these manual curation steps and devise strategies to automate these steps. The benefits of such future work would likely be two-fold: it would increase the reproducibility of these methods, and it could yield ensemble methods that are more successful in identifying optimal compounds for a given set of targets and anti-targets.

The predominant limitation of this challenge was a relatively small participant pool, which likely resulted in constrained space of methodology. Remarkably, the top-performing methods predicted the identical compound (ZINC98209221) as the best candidate for solving Problem 1, which is surprising given the size of the ZINC15 compound library. Furthermore, previous DREAM Challenges have consistently demonstrated that a variety of methodologic approaches can perform well in many different applications; therefore, it may be the case that methodologic approaches beyond neural networks may indeed be applicable to the problems defined in this challenge, but that the implementation may need to be altered to be successful.

The other key limitation of this study was the difficulty in translating computationally predicted best binders to activity in an *in vivo* system. We tested the top prediction from Problem

 

1 in the Drosophila RET$^{M955}$ model system of RET-driven cancer but did not observe bioactivity. There are multiple possible explanations for this, such as the standard ADME challenges faced in drug development, as well as the differences between the fly and human targets and networks [23,41]. To ameliorate this, improvements could include increasing the scale of experimentally tested compounds to identify a larger number of best binders, performing more stringent computational filtering to eliminate compounds with poorer predicted bioavailability characteristics, and performing standard medicinal chemistry and structure-activity relationship studies to identify analogues with better *in vivo* performance.

In the Results, we described a proof-of-concept approach to use these methods to identify molecular candidates for future experimental validation. By leveraging a "wisdom of the crowds" approach, we identified consensus Problem 1 predictions of the top-performing models by identifying the nexus of chemical space predicted by these methods. We subsequently evaluated these molecules using computationally ADME metrics to facilitate future prioritization of these molecules for medicinal chemistry and *in vivo* screening. With this in mind, employing a tandem approach that leverages ensembled computational models to identify a general region of chemical space with an appropriate polypharmacological profile, followed by more traditional drug optimization approaches, may be a worthwhile future avenue to accelerate the development of multi-targeting drugs.

## Materials and methods

### Challenge infrastructure

Multi-targeting Drug DREAM Challenge participants were required to submit a prediction file following a pre-formatted YAML template that included each team's top 5 compound predictions for each of the two problems (RET$^{M955T1}$- and TAU$^{R406W}$-mediated phenotypes) (Fig 1). Each prediction included a compound name, vendor name and ID, a methods description, a rationale explaining why the approach was innovative, how the approach is generalizable, and why top predicted compounds are chemically novel. Participants were restricted to compounds that were available through a vendor listed on ZINC15 or other vendors for no more than $250 per mg. The full set of predictions and methods descriptions provided by the Multi-targeting Drug DREAM Challenge community is available as Supplementary Methods.

### Top-performing algorithms

Detailed descriptions of all methods, including the top performing methods (Zhaoping Xiong and DMIS-MTD) are available in the Supplementary Methods file. Additional resources (e.g. final protein embeddings used) for each method can be obtained as described in the *Code and Data Availability section*.

### In vitro testing

Compounds were purchased and delivered to a contract research organization for experimental testing of binding to the target and anti-target kinases defined for Problems 1 (Fig 2) and 2 (Fig 4). Binding was evaluated using the DiscoverX KinomeScan profiling service. In brief, this method evaluated the ability of the predicted compounds to compete with a bead-immobilized ligand for the active sites of DNA-tagged kinases from Problem 1 and Problem 2. Kinases bound to the immobilized ligand were exposed to a broad range of concentrations of the prediction compounds (11 doses, 3-fold serial dilutions starting at 30 μM of compound). At each dose, the amount of drug bound to kinase was measured by eluting the non-bead-bound DNA-tagged kinase and performing quantification by quantitative polymerase chain reaction (qPCR).

### Scoring algorithm

Submissions were assessed using a point-based approach (Table 1). Points were awarded for meeting multiple criteria, in which "binding" of a pro-target was a positive hit at 10 μM and "non-binding" of an anti-target was defined as a negative result at 30 μM (S1 and S2 Tables).

For problem 1, each of the compounds submitted were awarded points for binding RET [M918T], BRAF, SRC, and/or S6K. Compounds were also awarded points for avoiding MKNK1, TTK, ERK8, PDK1, and/or PAK3. For problem 2: E, each of the compounds submitted were awarded points for binding AURKA, FGFR1, LKB1, and PAK1 and avoiding the anti-targets PAK3, MAP3K7 (TAK1), and PIK3CA. In addition, points were awarded for predictions for meeting other criteria including: novelty (ECFP6-based Tanimoto coefficient <0.4 against all other ChEMBL compounds known to be active against these targets), ability to patent (<2 hits in SciFinder database), and drug-like features (adheres to ¾ of Lipinski's rule of 5). In addition, for Problem 2, predictions predicted to enter the central nervous system (polar surface area <75) were awarded additional points.

### Chemical similarity calculations

Chemical similarity calculations and visualizations were performed in the R programming language using rcdk, fingerprint, pheatmap and visNetwork packages. SMILES strings for each structure were provided by each team, and then converted to a binary fingerprint using the "standard" method provided by the fingerprint package (a 1024-bit "path based, hashed fingerprint"). These fingerprints were then used to generate pairwise similarity matrices using the fingerprint::fp.sim. matrix with the method parameter set to 'tanimoto'. Full R notebooks describing these analyses can be found at https://github.com/Sage-Bionetworks-Challenges/Multi-Target-Challenge.

### Ensemble method

Ensemble predictions were performed by generating lists of top predictions for each top-performing team and assessing structural similarity as described above. We then used the SwissADME [42] webapp to calculate the number of Lipinski violations, Ghose violations, Veber violations, Egan violations, Muegge violations, PAINS alerts, Brenk alerts, and Lead-likeness violations (S5 Table). For each compound, we assigned a score of 1 for each value without a violation or alert, or a score of 0 otherwise. We then summed these values to create a composite score where a larger score indicates fewer applicable violations/alerts to prioritize compounds for in vivo screening.

### In vivo validation

Female *UAS-RET*$^{M955T}$ virgins were mated to *ptc-GAL4* males to generate *ptc>RET*$^{M955T}$ progeny. Progeny were grown at 25°C and compounds were tested by feeding larvae at three doses: 10 μM, 5 μM, and 1 μM final food concentration. DMSO alone served as a negative control and the experimental compounds AD80 and APS6-45 [23,37] served as positive controls. Using survival to adulthood (eclosure), each condition was done in quadruplicate, with 41–110 animals tested for each drug concentration.

## Supporting information

**S1 Methods. Supplementary description of the methods used to generate the predictions submitted by the Multi-targeting Drug DREAM Challenge community.**
(DOCX)

**S1 Fig. Chemical similarity comparison of all submitted Problem 1 and Problem 2 solutions.**
(TIFF)

**S2 Fig.** (A) Similarity heatmap of top predicted compounds for Problem 1 provided by the top-performing teams. Columns correspond to predictions from Zhaoping Xiong, while rows correspond to predictions from DMIS-MTD. The majority of compounds are relatively dissimilar from one another. (B) The similarity matrix was converted into a network, where nodes are individual predicted compounds and edges encode compound-compound Tanimoto similarities. Edge thickness represents similarity (thicker edges = greater similarity). Edges representing similarity below 0.4 were filtered out, and cluster subnetworks were identified (with each color representing an individual subnetwork).
(TIFF)

**S3 Fig. Recapitulating Zhaoping Xiong's three-class classification model used for screening.** (A) The distribution of pIC50; (B) The distribution of classes; (C) The accuracy of the model across train, valid and test sets.
(TIFF)

**S4 Fig. RET2B and control *D. melanogaster* were treated with DMSO (negative control), two candidate RET-targeting compounds identified using top performing methods (ZINC4020 and ZINC9820) and two positive control multi-targeting RET inhibitors previously demonstrated to rescue the RET2B model (AD80 [37], APS6-45 [23]).** Bars show the mean percent survival and error bars show the standard deviation of four replicates per condition. A Mann-Whitney test was used to assess the significance of changes to percent survival in the varying conditions. The RET2B Adult panel (bottom right) indicates that higher concentrations of AD80 or APS6-45 significantly ($p<0.05$) rescue the RET2B model, while the two candidate ZINC molecules have a minimal and non statistically-significant effect on percent survival to adulthood.
(TIFF)

**S1 Table. Problem 1 Results.** "Submitter or Team" indicates which team was scored. "Chemical ID" indicates the ZINC or MolPort identifier for each predicted compound. "InChIKey" indicates the InChIKey for each predicted compound. Columns 4–12 indicate the experimentally-validated binding constant ($K_d$, nanomolar) for the pro-target or anti-target listed in the column header (>30000 nM indicates it was above the detection limit for the assay). Columns 13–16 indicate the status of the compound with respect to the challenge rules for novelty, drug-like properties, and likelihood to pass the CNS as defined in the challenge scoring rules. The final column indicates the number of points the prediction was given based on the defined scoring criteria.
(XLSX)

**S2 Table. Problem 2 Results.** "Submitter or Team" indicates which team was scored. "Chemical ID" indicates the ZINC or MolPort identifier for each predicted compound. "InChIKey" indicates the InChIKey for each predicted compound. Columns 4–10 indicate the experimentally-validated binding constant ($K_d$, nanomolar) for the pro-target or anti-target listed in the column header (>30000 nM indicates it was above the detection limit for the assay). Columns 11–14 indicate the status of the compound with respect to the challenge rules for novelty, drug-like properties, and likelihood to pass the CNS as defined in the challenge scoring rules. The final column indicates the number of points the prediction was given based on the defined scoring criteria.
(XLSX)

**S3 Table. Zhaoping Xiong's method predictive performance on regression models.** A summary of the performance of various model architectures tested by top performer Zhaoping Xiong using root-mean-square-error calculated using true pIC50 values.
(XLSX)

**S4 Table. Top Problem 1 predictions fail to rescue the RET2B fly model.** The RET2B model was treated with two positive controls expected to rescue this model (AD80 and APS6-45), a vehicle control (DMSO), and two test compounds (ZINC4090, ZINC9820) predicted by the challenge results to rescue the RET2B model. "tx": the treatment (compound and concentration) used; "genotype": WT or RET2B flies; stage: class of counted animal—adult or pupae; "mean_percent_surv": the mean percent survival across 4 replicates; "std_dev_survival": the standard deviation of the percent survival across 4 replicates.
(XLSX)

**S5 Table.** *in silico* **chemical modeling of Problem 1 predictions from top performers.** Additional hits from top performing teams were characterized using SwissADME to identify compounds with preferred absorption, distribution, metabolism, excretion, and other medicinal chemistry properties.
(XLSX)

## Acknowledgments

The members of the Multi-targeting Drug DREAM Challenge Community in addition to the article byline include: Joerg Kurt Wegner, Janssen Pharmaceuticals; Huub Henkelsma, Gerard JP van Westen, Leiden Academic Center for Drug Research; Brandon Bongers, Leiden University; Lindsey Burggraaff; Leiden University, Jesper Van Engelen; Leiden University, Xuhan Liu, Leiden University; Xuhan Liu; Marina Gorostiola Gonzalez; Marvin Steijaert; Hugo Gutiérrez de Teran, Uppsala University; Holger Hoos; Anthe Janssen; Andre Falcao, University of Lisbon; Samina Kausar, University of Lisbon; Miguel Rocha, Centre Biological Engineering; Delora Baptista; Jorge Miguel Lourenço Ferreira, University of Minho; Jinhyuk Lee; Hwisang Jeon; Miyoung Ko; Donghyeon Park, Korea University; Gregory Koytiger, Immuneering Corporation; Team Stratified (4 anonymous members); Masahiro Mochizuki, DeNA Co., Ltd.; David Koes, University of Pittsburgh; Huiyuan Chen; Xengie Doan, Sage Bionetworks; Nasim Sanati, Sage Bionetworks.

## Author Contributions

**Conceptualization:** Kristen K. Dang, Alex Tropsha, Heather A. Carlson, Ruben Abagyan, Justin Guinney, Avner Schlessinger, Ross Cagan.

**Data curation:** Robert J. Allaway, Xindi Guo.

**Formal analysis:** Zhaoping Xiong, Minji Jeon, Robert J. Allaway.

**Funding acquisition:** Ruben Abagyan, Justin Guinney, Avner Schlessinger, Ross Cagan.

**Investigation:** Zhaoping Xiong, Minji Jeon, Robert J. Allaway, Mingyue Zheng, Xindi Guo, Alex Tropsha, Heather A. Carlson, Ruben Abagyan, Justin Guinney, Avner Schlessinger, Ross Cagan.

**Methodology:** Minji Jeon, Donghyeon Park, Jinhyuk Lee, Hwisang Jeon, Miyoung Ko, Hualiang Jiang, Aik Choon Tan, Xindi Guo, Alex Tropsha, Heather A. Carlson, Ruben Abagyan, Justin Guinney, Avner Schlessinger, Ross Cagan.

**Project administration:** Justin Guinney, Ross Cagan.

**Resources:** Xindi Guo, Tirtha K. Das.

**Software:** Zhaoping Xiong, Minji Jeon, Robert J. Allaway, Donghyeon Park, Jinhyuk Lee, Hwisang Jeon, Miyoung Ko, Hualiang Jiang, Mingyue Zheng, Aik Choon Tan, Xindi Guo.

**Supervision:** Justin Guinney, Avner Schlessinger, Ross Cagan.

**Validation:** Zhaoping Xiong, Minji Jeon, Robert J. Allaway.

**Visualization:** Zhaoping Xiong, Minji Jeon, Robert J. Allaway.

**Writing – original draft:** Zhaoping Xiong, Minji Jeon, Robert J. Allaway, Jaewoo Kang, Donghyeon Park, Jinhyuk Lee, Hwisang Jeon, Miyoung Ko, Hualiang Jiang, Mingyue Zheng, Aik Choon Tan, Kristen K. Dang, Alex Tropsha, Heather A. Carlson, Ruben Abagyan, Justin Guinney, Avner Schlessinger, Ross Cagan.

**Writing – review & editing:** Zhaoping Xiong, Minji Jeon, Robert J. Allaway, Jaewoo Kang, Donghyeon Park, Jinhyuk Lee, Hwisang Jeon, Miyoung Ko, Hualiang Jiang, Mingyue Zheng, Aik Choon Tan, Xindi Guo, Kristen K. Dang, Alex Tropsha, Chana Hecht, Tirtha K. Das, Heather A. Carlson, Ruben Abagyan, Justin Guinney, Avner Schlessinger, Ross Cagan.

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
