## [Decision Letter · Decision Letter 0]

9 Apr 2021

Dear Dr. Cagan,

Thank you very much for submitting your manuscript "Crowdsourced identification of multi-target kinase inhibitors for RET- and TAU- based disease: the Multi-Targeting Drug DREAM Challenge" for consideration at PLOS Computational Biology.

As with all papers reviewed by the journal, your manuscript was reviewed by members of the editorial board and by several independent reviewers. In light of the reviews (below this email), we would like to invite the resubmission of a significantly-revised version that takes into account the reviewers' comments.

We cannot make any decision about publication until we have seen the revised manuscript and your response to the reviewers' comments. Your revised manuscript is also likely to be sent to reviewers for further evaluation.

Sincerely,

Edwin Wang

Benchmarking Editor

PLOS Computational Biology

Edwin Wang

Benchmarking Editor

PLOS Computational Biology

Reviewer's Responses to Questions

**Comments to the Authors:**

Reviewer #1: In this manuscript, the authors reported the main results of the Multi-Targeting Drug DREAM Challenge, in developing computational methods for prediction of multi-target kinase inhibitors to treat RET-based cancer models and TAU-based neurodegenerative disease model in Drosophila. This Challenge was not organized in a very intelligent manner, and the results might not be so striking. I think the methods developed in this Challenge will be useful, however, only considering static features of proteins or small molecules might not ensure to reach a promising accuracy in reality. Since this Challenge has been finished, and indeed all participants have tried their best on method development, I think this paper can be accepted, after a minor but essential revision. I hope the authors can organize their future Challenge in a more sophisticated manner, and developed methods should touch the real fundamental problems. My major concerns are below:

1. The basic rationales of the Challenge were established on hypotheses. In Problem 1, the aim was prediction of small molecules that efficiently bound RET, BRAF, SRC and S6K, and didn’t bind MKNK1, TTK, ERK8, PDK1, and PAK3. In Problem 2, the aim was prediction of small molecules that efficiently bound AURKA, PAK1, FGFR1, and STK11, and didn’t bind PAK3, MAP3K7, and PIK3CA. 1) I do not think the concepts of pro-targets and anti-targets have been widely adopted by academic community, and targeting pro-targets but not anti-targets might not be the central dogma in drug design. 2) The basic hypothesis behind the Challenge is that if a small molecule can efficiently bind pro-targets but not anti-targets, corresponding diseases will be efficiently cured. I do not think this hypothesis is right. The identified small molecules might have additional side effects, and might be toxic in humans. 3) The second hypothesis behind the Challenge is that experimental in Drosophila can be used in humans. I also do not agree with this point. Even experiments in mice might make nonsense in humans. Human diseases are very complicated, and I do not think fly model can be really useful. Since the Challenge has been finalized, I hope the authors can be more cautious on organizing such competition in the future. A good Challenge cannot be founded on sand piles.

2. In the biological aspect, drugs that can efficiently treat the human diseases but not one or multiple proteins will be more preferred. If a drug can indeed block some proteins but cannot efficiently treat the disease, I do not think such drugs can be useful. Thus, an additional assay should be adopted to validate whether selected drugs can efficiently inhibit RET- and TAU-based models. If the authors cannot do such experiments, they must demonstrated why selected drugs targeting some proteins but not others can be really useful in disease treatment.

3. Prediction of multi-protein targeting inhibitors is an interesting topic. However, I do not think using static features derived from proteins or small molecules is enough to reach a promising accuracy. Image-base screening of small molecules using an appropriate readout might also be efficient. Thus, combination of multi-modal data types might be more useful.

4. Method 1 and 2 should be described with more details. Features derived from proteins or small molecules should be listed and interpreted in an Excel table. Used parameters in neural networks should be present to enable a full reproducibility. Please give a formal names of Zhaoping Xiong’s method.

Taken together, I think the paper can be accepted, although the Challenge did not touch a very important problem. In the past years, a number of such Challenges have been organized. None of them really resolved any important problems.

Reviewer #2: The authors described the development of the Multi-Targeting Drug DREAM Challenge for the identification of multi-target kinase inhibitors. This development is necessary because multi-targeting therapeutics have key advantages over single-targeting therapeutics, but it is difficult to evaluate which molecules have desirable target profiles. The authors designed two problems and around ten teams submitted their results to this Challenge. The authors combined computational approaches and experiment validation to evaluate the submitted drugs. The models of two teams with the top scores were shared with the public through Github. This manuscript is well-structured. I recommend it for publication.

**Have all data underlying the figures and results presented in the manuscript been provided?**

Reviewer #1: **No: **Features used in descrbed methods are not available and should be provided in the revision.

PLOS authors have the option to publish the peer review history of their article (what does this mean?). If published, this will include your full peer review and any attached files.

Reviewer #1: No

Reviewer #2: No

**Have the authors made all data and (if applicable) computational code underlying the findings in their manuscript fully available?**

Reviewer #2: Yes
---

## [Decision Letter · Decision Letter 1]

23 Jul 2021

Dear Dr. Cagan,

We are pleased to inform you that your manuscript 'Crowdsourced identification of multi-target kinase inhibitors for RET- and TAU- based disease: the Multi-Targeting Drug DREAM Challenge' has been provisionally accepted for publication in PLOS Computational Biology.

Best regards,

Edwin Wang

Benchmarking Editor

PLOS Computational Biology

Edwin Wang

Benchmarking Editor

PLOS Computational Biology

Reviewer's Responses to Questions

**Comments to the Authors:**

Reviewer #1: The authors carefully revised the manuscript, and addressed all my concerns. The current form is ready for publication.

**Have the authors made all data and (if applicable) computational code underlying the findings in their manuscript fully available?**

Reviewer #1: Yes

PLOS authors have the option to publish the peer review history of their article (what does this mean?). If published, this will include your full peer review and any attached files.

Reviewer #1: **Yes: **Yu Xue

---

## [Editor Report · Acceptance letter]

8 Sep 2021

PCOMPBIOL-D-21-00308R1 

Crowdsourced identification of multi-target kinase inhibitors for RET- and TAU- based disease: the Multi-Targeting Drug DREAM Challenge

Dear Dr Cagan,

I am pleased to inform you that your manuscript has been formally accepted for publication in PLOS Computational Biology. Your manuscript is now with our production department and you will be notified of the publication date in due course.

With kind regards,

Katalin Szabo
